# Evaluation of Sheet-Coated Composite Waterproofing Joint Types by Analysis of Tensile Strength Change and Concrete Displacement Resistance Testing under Environmental Degradation

**DOI:** 10.3390/ma13092120

**Published:** 2020-05-03

**Authors:** Chang-Pyo Chung, Su-Young Choi, Dong-Bum Kim, Wan-Goo Park, Byoung-Il Kim, Sang-Keun Oh

**Affiliations:** 1Program of Architecture of Convergence Institute of Biomedical Engineering and Biomaterials of Graduate School, Seoul National University of Science and Technology, 232 Gongneung-ro, Nowon-gu, Seoul 01811, Korea; yc4422@hanmail.net (C.-P.C.); db2128@naver.com (D.-B.K.); 2New Material & Convergence Laboratory Co., Ltd, 232 Gongneung-ro, Nowon-gu, Seoul 01811, Korea; csyoung777@gmail.com; 3Department of Architecture of Graduate School, Seoul National University of Science & Technology, 232 Gongneung-ro, Nowon-gu, Seoul 01811, Korea; dhdhkdrn@naver.com; 4School of Architecture, Seoul National University of Science & Technology, 232 Gongneung-ro, Nowon-gu, Seoul 01811, Korea; bikim@seoultech.ac.kr

**Keywords:** sheet-coated composite waterproofing, zero-span tensile stress, tensile strength, waterproofing joints, overlap joint, environmental degradation

## Abstract

Sheet-coated composite waterproofing (SCCW) have been developed to overcome the natural weakness of singly-ply coating or sheet waterproofing systems for roofing, but there are currently multiple types of SCCW joints. Conventional standard tensile strength testing results show that all SCCW joint types seem to pass the minimum requirement and current selection of SCCW type is dictated on the principle of ‘higher tensile strength is better’, but it has not been experimentally studied as to which type is the optimal to respond to environmental degradation while under the effect of zero-span tensile stress occurring during concrete joint displacement. In this study, five types of SCCW joints were tested: Overlap Bond (OB) type, Overlap Heated-Air Welding (OH) type, Butt Joint I Type (BI), Butt Joint T Type (BT), and Butt Joint Separation Movement Type (BS). These types of joint designs were subjected to Alkali, NaCl, and H_2_SO_4_ exposure, and temperature change (60 °C and −20 °C) for determining changes to tensile strength in the joint section. Tensile strength change results are compared to joint displacement resistance test results of specimens that were treated with chemical and temperature degradation. With the exception of chemical exposure conditioning, the Overlap type joints generally had higher tensile strength compared to the butt joint types, but joint displacement test results showed the opposite results, suggesting that complex joints found in SCCW require new evaluation method for quality assessment.

## 1. Introduction; Background

Until the early 2000s, most roofing protection consisted of using either a single-ply coat or sheet material comprised of rubber asphalt and urethane or asphalt and synthetic polymer–based sheet material [1,2,3]. Common problems with these types of waterproofing systems involved peel-off of the waterproofing layer, delamination, fractures localizing in the concrete cracks/joints. A proposed alternative to single-ply waterproofing was developed by utilizing the advantages of both sheet and coating methods to develop a “sheet-coated composite waterproofing (SCCW) method,” where both coating and waterproof sheet materials are installed together to form a waterproofing protection in roofing [4,5].

The SCCW systems are generally installed by first applying a sheet layer, followed by a coating layer at the joint or overlap section of the sheets for reinforcing support [6]. This method is intended to secure the stability of the joint by blocking exposure to external deterioration environmental factors through reinforcement against defects such as tearing, lifting, peel-off, and cracking at the joint [7,8,9]. However, SCCW joint defect problems persist in some sites where the SCCW method is applied as the degradation mechanism, mostly involving concrete displacement induced zero-span tensile stress, is still not fully understood. Depending on how the composite joint in the SCCW is formed, the waterproofing system can still be susceptible to leakage path forming at the interface of the sheet overlap or the concrete surface adhesion [10]. The following subsections will introduce the types of SCCW materials and joint types and methods currently being used in the market, and discussions on the problems with the evaluation methodology of SCCW joints, and the proposed direction of the study for the purpose of resolving these problems.

### 1.1. SCCW Joint Types and Methods

SCCW joint types include two broad categories: overlap methods and butt joint methods. For the overlap methods there are two types: Overlap Bond Type (OB) and Overlap Heated-Air Welding Type (OH). For the butt joint methods, there are three types: Butt Joint I Type (BI), Butt Joint T Type (BT), and Butt Joint Separation Movement Type (BS). The SCCW specimens tested in this study were installed with the same installation methods (as close as possible to the prescribed methods in the product specification) shown in Table 1 below [11].

### 1.2. Degradation Mechanism

For single-ply waterproofing methods, joint section failure can occur in the form of bonding failure, peel-off, gap formation under the joint overlap interface, and others [11]. There are multiple causes for joint failure due to environmental degradation (wind, settlement, introduction of chemical substance, thermal variation, etc.). In this paper, one degradation, and perhaps most important when it comes to SCCW type joints, is focused: zero-span tensile stress occurring during the displacement of concrete joints or cracks. In Japan, studies of waterproofing membrane application include as part of the Japan Society for Civil Engineering (JSCE) Standard (JSCE 2002) zero-span tensile test as a required performance property [12]. Their studies reveal the localization phenomenon of waterproofing membrane cracking due to the displacement of concrete joint. In the case of fluid-applied/coating/cementitious type waterproofing membrane systems (single-ply waterproofing methods), the displacement in the cracking of the membrane structure causes the stress to apply directly to the cohesive bond of the waterproofing layer and adhesion force on the concrete surface. In Figure 1 below is displayed the zero-span tensile stress mechanism applying to different types of waterproofing membranes adhered over concrete crack/joint.

This subject is particularly important with regards to securing proper quality with SCCW systems. During installation of waterproofing in roofing, conventional waterproofing requires forming some sort of joint (either overlap or T or I-type joint) to waterproof a large surface area. If the waterproofing membranes are not sufficiently cured, or if the concrete surfaces are not cleaned prior to installation, gaps can form along the interface of these overlap joints. Thicker waterproofing sheets have higher elastic modulus, and its elastic recovery applies additional straining force on the adhesion [13]. Refer to Table 2 below for the illustration of gap formation in joints.

These gaps at the overlap section interface become highly susceptible to failure in the form of water leakage path formation during zero-span tensile stress caused by concrete joint/crack displacement. In most cases, overlap joint adhesion is a highly technical procedure and number of sources indicate that this has been a topic of important discussion [14]. One of the main philosophies as to why SCCW was developed is to reduce the reliance on skilled workmanship and simplify the installation procedure by introducing a supposed fail-proof waterproofing membrane joint precisely due to reasons like this. In recent years, while SCCW was used in Korea, cases of overlap/joint failure due to concrete displacement have reduced, but there are still cases where failures occur due to environmental degradation [14]. A proposed solution to this persisting problem is to propose a new evaluation method that can simulate the a realistic zero-span tensile stress conditioning after subjection to environmental degradation to assess the SCCW systems. To discuss the necessity of this new evaluation method, first the short comings of conventional international standard testing method need to be discussed.

### 1.3. Evaluation Criteria for SCCW Performance; Resistance to Concrete Displacement

Some of the representative international standard test methods with regards to evaluation of waterproofing membrane joint system evaluation under concrete displacement (substrate joint/crack movement) can be found in British Standard (BS EN 12316-2:2013), American Standard Tests and Materials (ASTM D5849 and ASTM C1305) and Korean Standard (KS F 4911) [15]. For these standards, interpretation of results is based on bonding failure, bridging failure, or tear resistance and the strength value is recorded and types of failure is visually observed and recorded. The philosophy of interpretation in these cases where tearing, surface defect, bridging failure, or bond breakage occurring under the required stress conditions (common with coating type materials such as urethane) is considered as an indication of low performance. Refer to Table 3 below lists specific reference materials from each national standard body.

The principles of minimal performance requirement in terms of tensile strength resistance value is a useful tool for comparative evaluation between different types of waterproofing materials and whether they can properly respond to the expected stress conditions in the structure. However, with materials types that pass the minimum requirements under these testing methods, leakage path formation and adhesion failure to concrete surface due to zero-span tension is not properly considered.

Leakage path formation is a commonly occurring problem with SCCW, especially after subjection to environmental degradation. As is explained in the above introduction, SCCW are comprised of multiple layers of different materials/reinforcing stiffener/bonding adhesives/release tapes/hot welding joints. SCCW can come with different types of joints (overlap and butt joint methods) based on the installation method, but not all joint types have the same resistance property level to zero-span tensile stress coupled with environmental degradation. With the conventional standard test methods, it is difficult to comparatively assess the concrete displacement resistance property of different SCCW joint types. To understand the structures of the different joint types, the following sections introduce the recently developed and used with SCCW methods [9].

### 1.4. Proposing a New Evaluation Regime for SCCW Roofing

One of the main problems with the employment of the SCCW systems is that the current quality standards and evaluation methods for waterproofing joints are oriented towards testing single-ply waterproof sheets. When tested under the conditions compliant in the existing methods (ASTM, BS EN, JIS, GB, or KS), SCCW systems are shown to perform significantly better than singly-ply systems, and SCCW systems have in fact been more successful than the single-ply counterparts. [8]. While it is true that workmanship and installation method, as well as the experience of the workers on-site ultimately determines the quality of the waterproofing system, understanding the correct mechanisms of degradation in the joint section is also significantly important. However, the existing test methods are not sufficient to completely simulate environmental degradation and concrete joint displacement at the same time, and the criteria of evaluation only observes changes to physical properties rather than waterproofing properties, and this can lead to an incomplete evaluation of SCCW system performance in roofing. To properly assess and select an optimal joint type for SCCW system, an evaluation method that goes beyond basic physical property testing method results, such as tensile strength measurement testing, is required. To demonstrate this point, tensile strength property, which is one of the main required performance property for determining quality assessment of waterproofing membranes, is compared with a concept that is to be called concrete displacement resistance performance of SCCW joints in underwater condition.

In this paper, conventional tensile strength test method results relevant to the SCCW joint section is first derived, and the SCCW joint tensile strength changes after environmental degradation is measured, and an evaluation based on standard criteria is performed. Afterward, a concrete displacement testing that simulates concrete displacement induced zero-span tensile stress is compared. For both testing, same environmental degradation conditioning is applied to the SCCW installed specimens, and the type trends of results are compared.

## 2. Experimental Plan and Method

### 2.1. Evaluation Method of SCCW Layer Joints (Overlap and Butt Joint Methods)

After the specimen is subjected to temperature and chemical deterioration: (1) the tensile bond performance is evaluated and (2) the concrete displacement resistance performance is evaluated to quantitatively confirm the performance of the joint as a result of the concrete displacement deterioration. The size of the SCCW specimens and test conditions for the tensile strength are compliant to the test method of “KS F 4917 Polymer-modified bitumen waterproofing sheet” of the Korean Industrial Standards (KS) [24]. The size and production method of the specimen for evaluating the performance corresponding to the concrete displacement were applied by test method (AIK-S-0001-2019) of the Architectural Institute of Korea (AIK) [25]. The general outline of the two tests above is shown in Figure 2.

### 2.2. Production of Specimens per Joint Types

In terms of the specimens to be tested, the joints were produced by using a polyurethane coating material, which is a thermoplastic polymer compound, and a polyvinyl chloride (PVC) waterproofing sheet, which are thermoplastics, among the materials most commonly used in the SCCW method. Table 4 shows the materials required to manufacture test pieces for different types of joints. For each testing criteria, 5 specimens for each joint type (25 tensile strength and 25 concrete displacement resistance test specimens) was prepared for each test conditioning (untreated, 2 types of temperature conditioning, and 3 types of chemical conditioning).

### 2.3. Joint Tensile Strength Test

Specimen preparation was conducted by the method compliant to KS A 0006. Each specimen were produced with a size of 100 mm width × 200 mm length (for illustration on the specimen joint assembly, refer to Table 1) and the length of the overlap/joint section was fixed to 10 mm. Completed specimens were left to cure at temperature condition of (20 ± 15) °C for 24 h and humidity level of (65 ± 20)%, and were then subjected to temperature and chemical deterioration treatment as shown in Table 2. After temperature and chemical deterioration for 168 h, the test pieces were removed from the deterioration condition and allowed to rest at room temperature for 1 h, and the tensile strength of the joint was measured. 

The tensile strength of each joint types was tested using an Universal Testing Machine (UTM, according to the ISO 9001:2008;QS-9000;CE specification, manufacturer: GAOXIN, Guangdong, China) apparatus (refer to Figure 3 for illustration) under temperature of (20 ± 3) °C and a relative humidity of (60 ± 5)% compliant to the conditions outlined in KS F 4917. After insertion of the test piece into the UTM, tensile force at a speed of 300 mm/min is applied until the specimen breaks, the tensile strength of the joint is read as the maximum load (N) and is expressed as the average value of 25 test pieces according to the equation below.
*T_B_* = *P_B_*/*W*(1)
where: *T_B_*: Tensile Bond Strength (MPa)*P_B_*: Maximum Load (N)*W:* Outside Insertion Surface (25 mm^2^)

The evaluation standard for the tensile strength results obtained from this testing are compliant to tensile performance criteria of “KS F 4917” that outlines more than 5.0 N/mm^2^ as a minimum value [23].

Details for the environmental degradation conditioning are outlined in Table 5 below, where a comparison between “standard (untreated) vs. environmental degradation (treated) condition.” Tensile strength test results and resistance testing to concrete displacement results under the untreated condition as standard (20 °C) for comparison with the degradation conditioning test results. For the degradation conditions, temperature variation (temperature of −20 °C and 60 °C), and chemical degradation (exposure to Alkali, NaCl, and H_2_SO_4_) was selected.

### 2.4. Concrete Displacement Resistance Test

#### 2.4.1. Concrete Displacement Resistance test: Specimen Preparation, Apparatus, and Method

In the concrete displacement test, a mortar mold (Water to Cement ratio = 0.6:1 in accordance to KS) of with diameter of 180 mm and a height of 130 mm is placed on top and bottom. A SCCW system is constructed so that a layered chip portion is formed around the mold joint. The waterproof layer is installed at 50 mm from the top of the upper specimen and the bottom of the lower specimen. At this time, the lower specimen has a hole with a diameter of 50 mm and a depth of 100 mm at the center, and the upper specimen is manufactured by inserting a steel T-shaped rod having a height of 150 mm and pouring mortar. A total of five specimens were manufactured for each type of joint. Figure 4 shows the specimens and test equipment based on the concrete displacement.

The concrete displacement inducing apparatus consists of a UTM (Universal Testing Machine) and a specially structured acrylic container. The test specimen is first installed in the acrylic container, which is then filled with water, and installed on to the UTM apparatus for concrete displacement resistance testing. Once installed, the upper mortar slab section of the specimen is pulled up and down vertically relative to the bottom slab section fixed to the bottom of the acrylic container. Refer to Figure 5 for details.

Below is a set of more detailed steps for installing the specimen onto the testing apparatus:(1)The test specimen is installed into the acrylic chamber apparatus by screwing the threaded conduit of the bottom mortar slab onto the metallic thread at the bottom of the chamber.(2)The container is filled with approximately 15 L of water. The specimen should be completely submerged in water.(3)The container is installed onto the UTM. The bottom connection is anchored with a rivet to the jig of the UTM. The upper mortar slab threaded conduit is then anchored to the upper jig of the UTM.(4)Testing procedure and conditioning is explained in Table 6 below.

#### 2.4.2. Interpretation of Result for Concrete Displacement Resistance Test

Cause of the leakage is classified into two broad categories: leakage path formation (due to adhesion failure or gap formation at the overlap/joint interface), or membrane tearing. A moisture sensor is installed at the leakage outlet at the bottom of the apparatus, where the leaked water is collected in a container. If continued streams of water begin to flow through the outlet, the testing is stopped and the interval during the displacement cycle at which the leakage occurred is recorded. To identify the cause of leakage, specimens are taken out of the chamber and the specimens undergo dissection (process illustrated in Figure 6 below). The waterproofing membrane is removed from the mortar substrate using a knife. During removal, completely exposed areas (in Figure 6d) below) are checked to see if heavy moisture is detected on the interface of the membrane and the mortar surface.

## 3. Test Results

### 3.1. Tensile Strength by Joint Type According to Temperature Deterioration Conditions

Table 7 and Figure 7 summarize the results of the tensile strength test for each joint type according to temperature degradation conditioning.

The results of tensile strength of the joints from highest to lowest could be outlined in the order of the following: OH > OB > BT > BS > BI. It was confirmed that the OH type showed the highest tensile strength under all deterioration conditions, and the BI type showed the lowest tensile strength. Overall, the overlap joints of OH (the heat-sealing integrated joint) and OB (the adhesive-bonded joint) showed higher tensile strength than the butt structure joints BI (butt joint with coating), BT (butt joint with reinforced stiffener), and BS (butt joint with release tape for separation movement). In the effect of temperature condition, in the 60 °C environment, a tendency to decrease in strength was observed compared to the 20 °C environment, and in the −20 °C environment, a tendency to increase in strength compared to the 20 °C environment was confirmed.

As the result, as shown in radar chart of Figure 5, after deterioration treatment at 20, 60, and −20 °C in all types of joints, it was confirmed that the tensile strengths of the SCCW joints were over 5.0 N/mm^2^, which is the quality (refer to value of triangle with dotted purple line in Figure 7) for tensile strength of single sheet joints in the standard specified in KS F 4917.

### 3.2. Tensile Strength by Joint Type According to Chemical Degradation Conditions

The results of the bonding tensile strength test for each joint type according to chemical degradation are summarized as shown in Table 8 and Figure 8.

As shown in Figure 8, the tensile strength test of the joint according to the chemical deterioration condition, the specimens subjected to the alkali and NaCl treatment conditions showed that the joint tensile strength descended in the order of OH > OB > BT > BS > BI. However, in the case of specimen subjected to the H_2_SO_4_ treatment condition, the order observed was OH > BT > OB > BS > BI.

As the result, as shown in radar chart of Figure 6, after the deterioration treatment of alkali, NaCl, and H_2_SO_4_ in all types of joints, it was confirmed that the tensile strength of the SCCW joints was over 5.0 N/mm^2^.

### 3.3. Concrete Displacement Testing Results

Concrete displacement resistance test results for the SCCW are summarized in Table 9 and Figure 9. Concrete displacement test results were determined by recording the interval during the displacement cycles at which leakage started to occur. OB averaged 208 number of times, OH 219 times, BI 516 times, BT 211 times, and BS 520 times. For the concrete displacement test, SCCW joint types with the highest to lowest concrete displacement resistance could be outlined in the following order: BS > BI > OH > BT > OB.

BI and BS joint types are able to withstand the stress generated in the zero-span tension applied to the waterproofing layer during concrete displacement owing to the fact that the sheet layers are not homogeneously joined, and the stress is dispersed through the flexible coating and release tapes in the respective joint types. In the case of OB, OH, and BT, as the joints formed between the sheet layers is held by a stronger bonding mechanism that provides the stronger tensile strength property of the joint types (as was shown in the results from Section 3.1 and Section 3.2). In this regard, the stress is more localized and focused at the joint movement point of the waterproofing layer due to zero-span tension. The results can be quantified as is shown in radar chart of Figure 8. As evidenced from the results of the green line below, the BI (520 number in Figure 8) and BS (516) were able to reach higher overall concrete displacement cycle number due to their nature of higher joint flexibility that can withstand zero-span tension, and OB (208), OH(219), and BT(211) reached lower concrete displacement cycle number due to their more stiffer joint property.

With regards to the results of the environmental degradation conditioning, it was shown that in all degradation conditioning in concrete displacement resistance testing, the performance deteriorated for all types of SCCW joints. However, the order in which the highest resistance performance found under untreated condition remained relatively the same, with BI and BS showing the highest resistance property, followed by OB, OH and BT. Refer to Figure 9 below for details.

## 4. Analysis and Discussion

Figure 10 and Figure 11 below summarize the tensile strength and the concrete displacement resistance results by joint type under temperature and chemical degradation conditions. In the tensile strength test for the joint, it was confirmed that the OH type showed the highest tensile strength under all deterioration conditions, and the BI type showed the lowest tensile strength. Overall, OB and OH showed higher tensile strength than BI, BT, and BS. Furthermore, it was confirmed through the above testing that all the joint types exceeded the tensile strength quality standard of 5.0 N/mm^2^. However, in contrast to the results of the tensile strength test, in the flexible type joints of BI and BS, where the tensile strength was relatively low, higher resistance to concrete displacement was measured than the SCCW types with stiffer joints of BT, OB, and OH where strength for resistance to concrete displacement was relatively high.

The results of this study show that high tensile strength at the joint can still indicate risks of ‘failure’ due to leakage, and this criterion is overlooked in existing international standard testing methods. Thus, high tensile strength alone cannot be said to indicate high resistance property to concrete displacement. In the case of KS F 4917, minimum tensile strength requirement does not serve as a clear indicator of waterproofing performance or integrity for SCCW systems. In this particular demonstration, a simplified correlation between tensile strength and concrete displacement resistance testing shows that current method of result interpretation found in international standards would not have been able to determine and compare the performance levels of the different type of SCCW joint types. Further studies with other parameters and variables compliant to different national standards will be required in order to establish a more objective evaluation method and parameter for SCCW using concrete displacement resistance testing.

## 5. Conclusions

A study was conducted to understand the degradation mechanism in the SCCW joints by: (1) calculating the changes to tensile strength in the five types of SCCW joints after temperature and chemical degradation conditioning and (2) conducting a testing to derive the resistance to concrete displacement for the five types of SCCW joints. The results are as follows:

(1) As a result of the tensile strength evaluation by joint type, in terms of all the five joint types’ performance after the degradation conditioning, it was confirmed overlapped joint type has a relatively higher tensile performance compared to the butt joint type.

(2) As to the influence of tensile performance according to the deterioration environmental conditions, it was confirmed that high temperature (60 °C) and acid (H_2_SO_4_) had a great influence on the deterioration of the joint’s performance, and tensile strength reduction of up to about 27% was observed in the Overlap Bond Type(OH). Based on this, it is estimated that sulfuric acid oxidizes the adhesive bonds of the joint to reduce the strength of the adhesive, resulting in a decrease in the tensile strength of the joint.

(3) For the concrete displacement resistance test, it was confirmed that the Butt Joint types (except for Butt Joint T type), had higher resistance performance than the Overlap joint types. The results of this test do not necessarily indicate that tensile strength property has an inverse relation to the concrete displacement resistance property. However, the comparison nevertheless indicates that the structure of the waterproofing joint types has a great influence on the expected waterproofing performance of the SCCW system in roofing, and conventional international testing methods may not suffice for a complete comparative evaluation.

## Figures and Tables

**Figure 1 materials-13-02120-f001:**
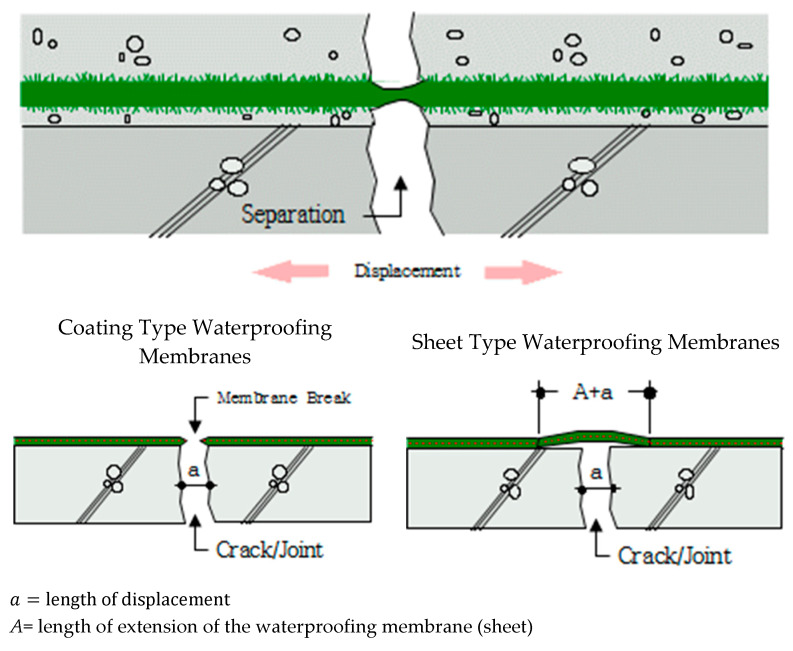
Zero-span tensile stress mechanism of waterproofing membrane on concrete joint.

**Figure 2 materials-13-02120-f002:**
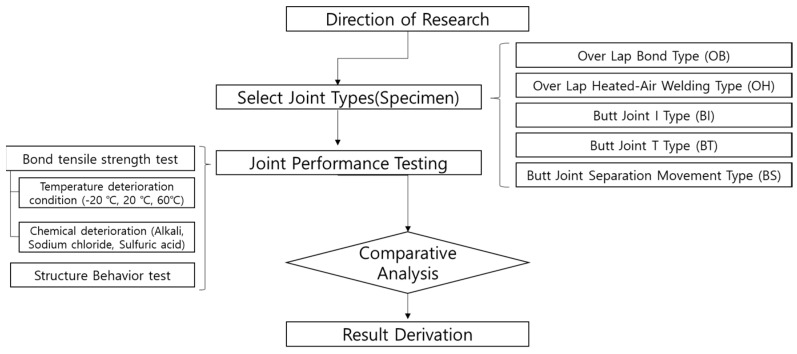
Flowchart of performance comparison and evaluation by joint type.

**Figure 3 materials-13-02120-f003:**
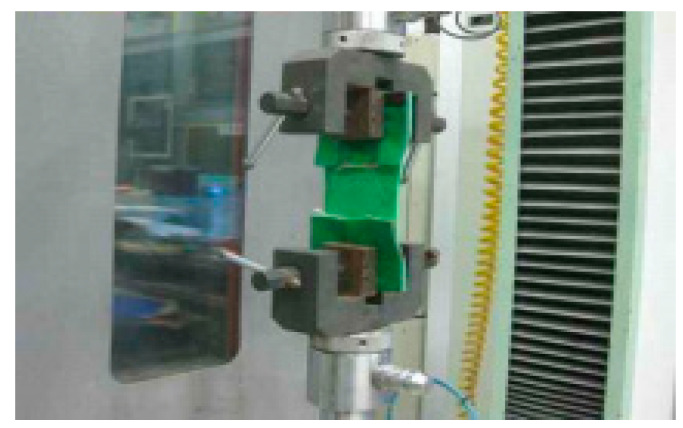
Joint tensile strength testing (using Universal Testing Machine (UTM)).

**Figure 4 materials-13-02120-f004:**
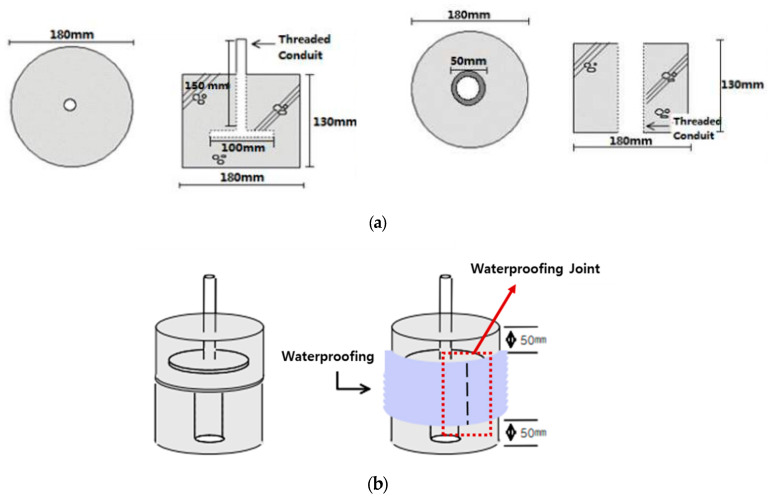
Concrete displacement resistance testing specimen: (**a**) mortar base and (**b**) SCCW component installation.

**Figure 5 materials-13-02120-f005:**
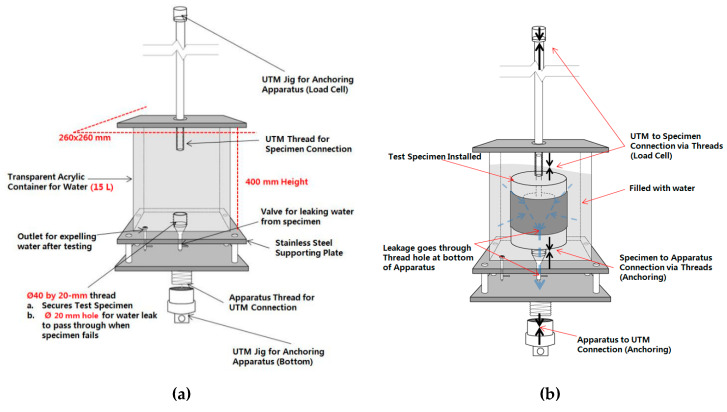
Concrete displacement resistance testing apparatus illustrated: (**a**) container and component dimensions and (**b**) specimen installation into the acrylic container apparatus.

**Figure 6 materials-13-02120-f006:**
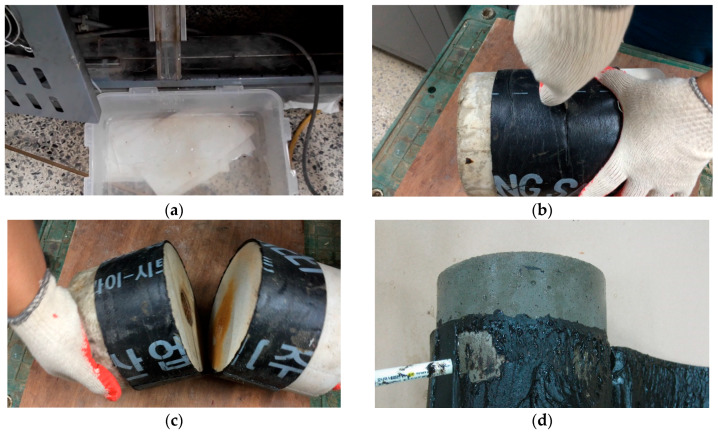
Waterproofing membrane evaluation after testing: (**a**) leakage detection, (**b**) incision to separate mortar slabs, (**c**) checking interior surface, and (**d**) analysis.

**Figure 7 materials-13-02120-f007:**
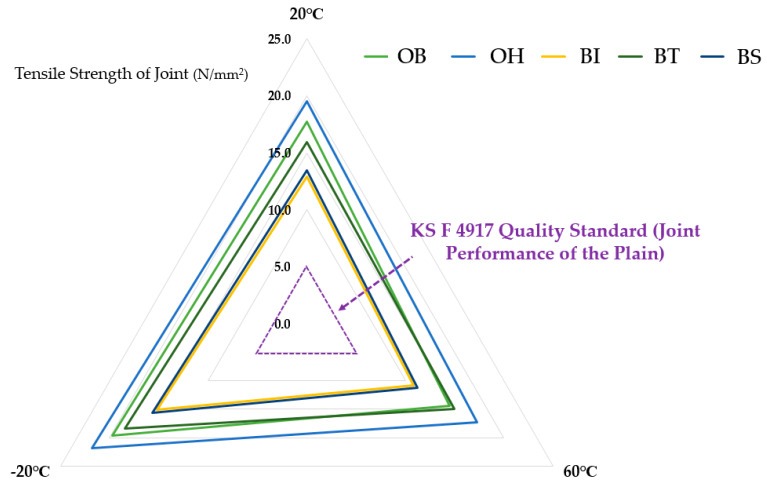
Joint tensile strength test results (temperature deterioration condition).

**Figure 8 materials-13-02120-f008:**
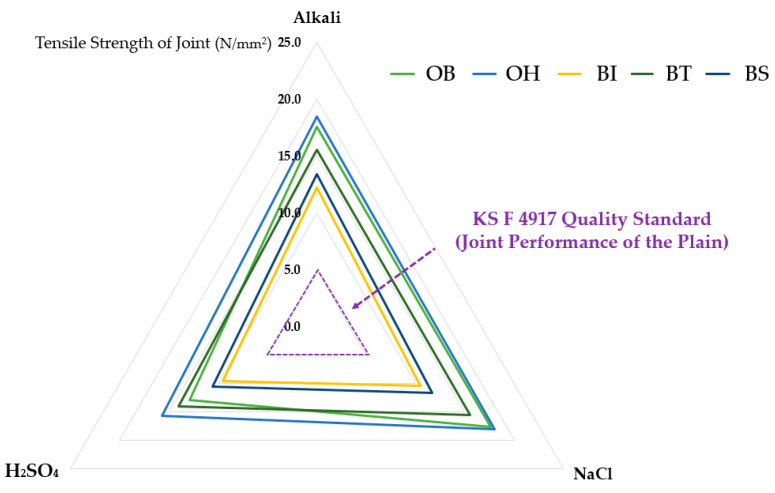
Test Results of Tensile Strength of Joint (Chemical Deterioration Condition).

**Figure 9 materials-13-02120-f009:**
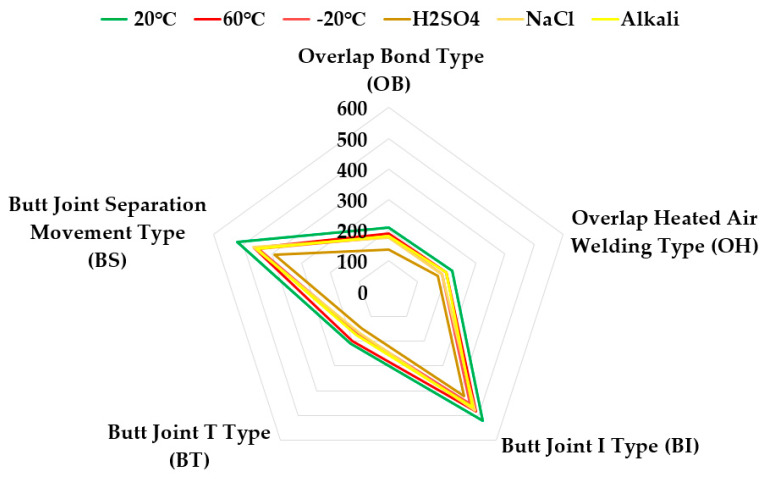
Concrete displacement test results.

**Figure 10 materials-13-02120-f010:**
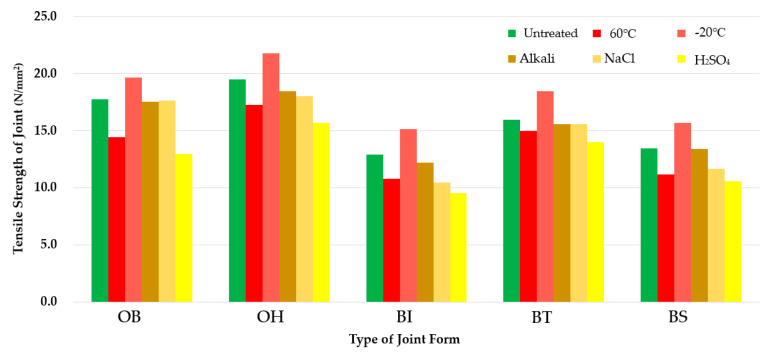
Tensile strength test results per SCCW joint types.

**Figure 11 materials-13-02120-f011:**
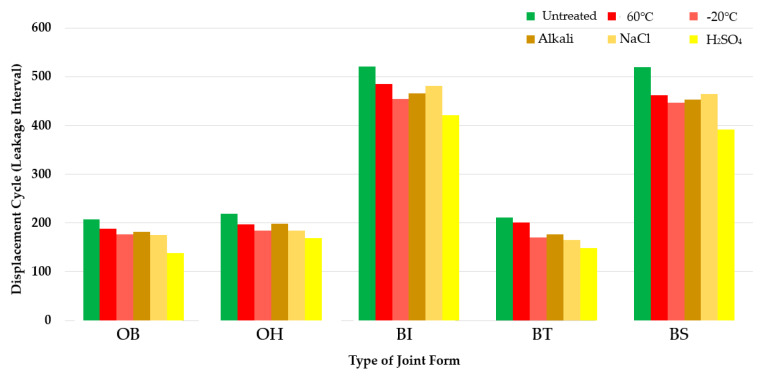
Concrete displacement results per SCCW joint types.

**Table 1 materials-13-02120-t001:** Sheet-coated composite waterproofing (SCCW) joint types.

Method	Type	Illustration (Image)		Installation Procedure
Overlap methods	Overlap Bond Type (OB)	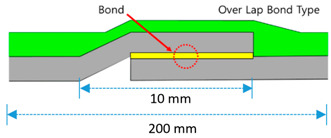	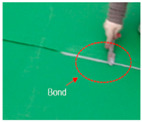	Gluing the sheet joint using adhesive
	Overlap Heated-Air Welding Type (OH)	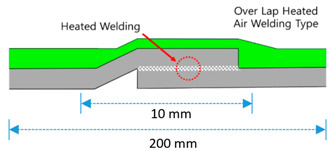	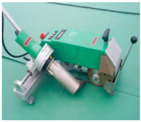	Sheet joints melted with a hot air fusion machine to adhere to each other
Butt Joint Methods	Butt Joint I Type (BI)	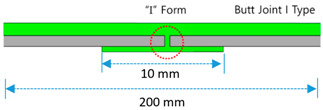	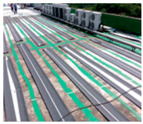	I-type joint butting with coating material
	Butt Joint T Type (BT)	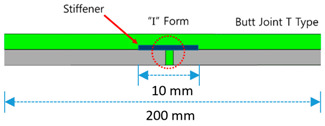	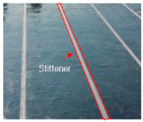	T-type joint using stiffener on butting part
	Butt Joint Separation Movement Type (BS)	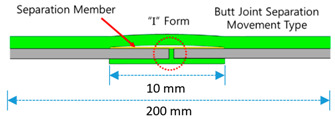	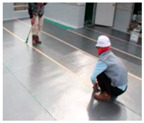	I-type joint using release tape (sliding member) on butting part

**Table 2 materials-13-02120-t002:** Illustration of Application Method for Waterproofing Membrane Sheet Overlap.

Standard Application Method for Waterproofing Sheet Overlap Joint (Single-ply and SCCW)	
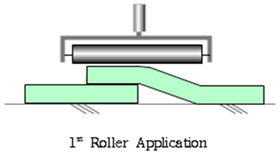	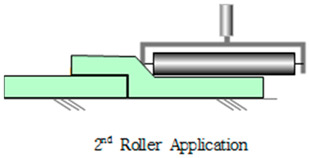
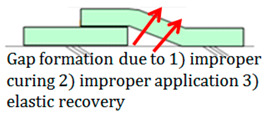	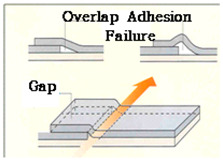
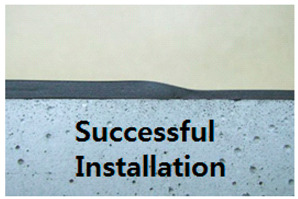	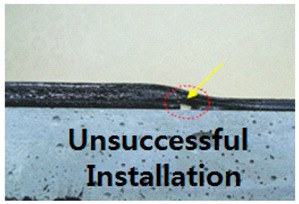

**Table 3 materials-13-02120-t003:** International Standards on Required Waterproofing Membrane Properties [16,17,18,19,20,21,22,23].

Nationality	Standard Code	Description/Relevance
US Standards (ASTM: American Society for Testing and Materials)	ASTM D 412 ASTM D 4637 ASTM C 1305 ASTM D 5405 ASTM D 4073 ASTM D 5849	Test methods, specifications, requirements related to tensile strength and properties relevant to concrete displacement resistance property. Concept on zero-span tensile stress (via different terminology) is present in each national standard in some way or form, especially in the case of ASTM (C1305). JSCE specifically mentions zero-span tensile stress testing as a requirement as well. However, interpretation of test results is based only on (1) minimum required strength value (*MPa* or N/mm^2^), or (2) visual observation and recording of the type of defect on the membrane after testing.
Korean Standards (KS: Korean Standard)	KS F 4917 KS F 4935	
Chinese Standards (GB: Guo Biao)	GB/T 328.8-2007	
Japanese Standards (JSCE: Japan Society of Civil Engineers, JIS: Japanese Industrial Standards)	JSCE K-532-1999 JIS 6013	
British Standards (BS: British Standards)	BS EN 12316-2:2013	

**Table 4 materials-13-02120-t004:** Material Information for Specimen Production.

Illustration	Title	Base Information	Application
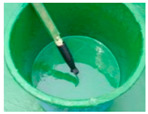	Coating Material	Urethane coating material mixed with polyurethane propolymer (55%–75%), perfluorooctyl ethanol (7%–14%), fluoropropane (10%–15%)	Base material
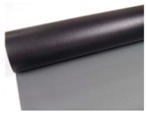	Sheet Material	PVC (100% new materials)	Base material
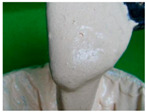	Glue	Subjects including epoxy resin (30%–40%), polyurethane (15%–25%), and calcium carbonate (25%–35%) and glue mixture with hardener containing polyamide resin (30%–60%) and isophorone diamine (20%–45%)	For overlap bond type
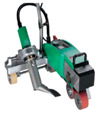	Hot Air Fusion Machine	Voltage: 230 V/400 V Power: 3680 W/5700 W Temperature: 100–600 °C Air Flow Range: 50%–100%	For overlap heated-air welding type
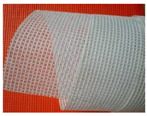	Reinforcement (stiffener)	Fiber glass	For T type buttress
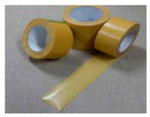	Release Tape (Separation Member)	Woven Fabric	For split movement buttress

**Table 5 materials-13-02120-t005:** Temperature and Chemical Deterioration Conditions as Environmental Treatment.

Criteria		Conditions			
**Standard Condition** **(Untreated)**		20 °C (168 h at rest)			
**Degradation** **Condition**	Temperature	−20 °C (168 h)		60 °C (168 h)	
	Chemical	Alkali (10% concentration, 168 h exposure)	NaCl (10% concentration, 168 h exposure)		H_2_SO_4_ (2% concentration, 168 h exposure)

Note: the test conditions outlined in this paper is for demonstration purposes only and are compliant to the specifications and regulations of Korea only. If this test is to be replicated elsewhere, different degradation conditioning and parameters can apply as well.

**Table 6 materials-13-02120-t006:** Concrete Displacement Test Method [19].

Concrete Displacement Test Method	Test Contents
Joint Displacement Testing Procedure	After the structure is filled with water in the tester where the test piece is installed, the concrete displacement is performed 100 times in each step in the water (20 ± 3 °C). The movement interval of the concrete displacement tester is set to 10 mm, and the movement distance is tested at a constant speed of 50 mm/min. The concrete displacement is repeated 100 times, and this is 1 cycle.
Underwater Condition (Untreated Standard Condition)	When testing the concrete displacement, the test piece is subjected to an underwater temperature of 20 ± 3 °C.
Degradation Condition	Specimens selected for degradation conditioned testing are placed inside thermal chamber for temperature variation and chemical substance mixed aqueous solution for chemical degradation treatment according to the same conditions outlined in Table 2 above. Once the degradation conditioning is completed, the specimens undergo the same joint displacement testing as the untreated version.

Note: This test procedure and conditions (recently made Korean standard, AIK-S-0001-2019) outlined in this paper is for demonstration purposes only and are compliant to the specifications and regulations of Korea only. If this test is to be replicated elsewhere, different degradation conditioning and parameters can apply as well.

**Table 7 materials-13-02120-t007:** Joint Tensile Strength Test Results (Temperature Deterioration Condition).

Temperature Condition	Criteria	Tensile Strength of Joint Types				
		OB	OH	BI	BT	BS
20 °C	Average (N/mm^2^) ^1)^	17.72	19.51	12.91	15.95	13.43
	Standard Deviation	0.73	0.73	0.53	0.57	0.51
	Coefficient of Variation (%)	4	4	4	4	4
60 °C	Average (N/mm^2^)	14.45	17.25	10.79	14.95	11.18
	Standard Deviation	0.44	0.56	0.38	0.48	0.48
	Coefficient of Variation (%)	3	3	3	3	4
−20 °C	Average (N/mm^2^)	19.68	21.78	15.14	18.44	15.66
	Standard Deviation	0.43	0.40	0.35	0.37	0.41
	Coefficient of Variation (%)	2	2	2	2	2

^1)^ Value obtained using Equation (1) from Section 2.3.

**Table 8 materials-13-02120-t008:** Joint Tensile Strength Results (Chemical Deterioration Condition).

Chemical Condition	Criteria	Tensile Strength of Joint Types				
		OB	OH	BI	BT	BS
Alkali	Average (N/mm^2^) ^1)^	17.52	18.48	12.20	15.56	13.41
	Standard Deviation	0.34	0.33	0.27	0.31	0.28
	Coefficient of Variation (%)	2	2	2	2	2
NaCl	Average (N/mm^2^)	17.62	18.04	10.48	15.55	11.66
	Standard Deviation	0.38	0.29	0.34	0.31	0.28
	Coefficient of Variation (%)	2	1	3	2	2
H2SO4	Average (N/mm^2^)	12.94	15.68	9.53	13.99	10.58
	Standard Deviation	0.61	0.25	0.32	0.33	0.29
	Coefficient of Variation (%)	5	2	3	24	27

^1)^ Value obtained using Equation (1) from Section 2.3.

**Table 9 materials-13-02120-t009:** Concrete Displacement Resistance for Waterproofing Joint Types According to Temperature and Chemical Deterioration.

Criteria		Specimens (Joint Types)				
		OB	OH	BI	BT	BS
Temperature	Measured Interval (20 °C) ^1)^	208	219	521	211	520
	Measured Interval (60 °C)	188	197	485	201	462
	Measured Interval (−20 °C)	177	184	454	170	447
Chemical Deterioration	Measured Interval (Alkali)	182	198	466	177	453
	Measured Interval (NaCl)	175	185	481	165	465
	Measured Interval (H_2_SO_4_)	138	169	421	148	392

^1)^ Untreated condition (standard).

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
