# Peer review of "Evaluation of Sheet-Coated Composite Waterproofing Joint Types by Analysis of Tensile Strength Change and Concrete Displacement Resistance Testing under Environmental Degradation"

_materials, 2020, doi:10.3390/ma13092120_

Round 1

Reviewer 1 Report

While the English language seems to contain largely correct grammatical structures, it is very difficult to glean the meaning as the language is too verbose and circumlocutory. It is impossible to proceed with the complete review when the meaning of wordings cannot be ascertained. A thorough revision with concise, succinct statements would greatly enhance the readability so that the review can be completed in its entirety. In addition, the following comments should be implemented.   

  1. SCCW should be renamed Sheet-Coated Composite Waterproofing instead of Sheet-Coating……
  2. In Table 1, the term Elongation is misleading, because Elongation usually means expanded length – original length. Another suitable term should be used to indicate the ratio of lengths after and before.
  3. Over-lap and Over Lap should be written as one word, overlap.
  4. The formula in Equation 1 is dimensionally incorrect. MPa=N/mm2≠N/mm. Please redefine.
  5. Relevant sections of English version of KS F 4917 can be provided.

Author Response

First and foremost, we would like to express our utmost gratitude for taking the time to read and comment on our manuscript. We understood from your feedback that our initial draft was rather difficult to understand due to poor quality in writing and arrangement. So we took the time to conduct further experimentation and revise the manuscript accordingly. Most importantly, we removed the latter part of the paper that relies on estimation for the results and replaced this section with experimental data on structural movement response testing (now called concrete displacement resistance) after environmental degradation. We hope the manuscript now reads more clearly and a complete review can be conducted. In the below, we have provided more specific response to your individual comments.

Starting comment;

While the English language seems to contain largely correct grammatical structures, it is very difficult to glean the meaning as the language is too verbose and circumlocutory. It is impossible to proceed with the complete review when the meaning of wordings cannot be ascertained. A thorough revision with concise, succinct statements would greatly enhance the readability so that the review can be completed in its entirety. In addition, the following comments should be implemented.  

Comments/Feedback

  1. SCCW should be renamed Sheet-Coated Composite Waterproofing instead of Sheet-Coating……

Response: SCCW name has been revised accordingly

  1. In Table 1, the term Elongation is misleading, because Elongation usually means expanded length – original length. Another suitable term should be used to indicate the ratio of lengths after and before.

Response: As you have mentioned, elongation is not a suitable term to describe this phenomenon. The term “displacement” has been used instead.

  1. Over-lap and Over Lap should be written as one word, overlap.

Response: Over-lap/Over Lap has been revised to overlap throughout the paper as suggested

  1. The formula in Equation 1 is dimensionally incorrect. MPa=N/mm2≠N/mm. Please redefine.

Response: Unit has been revised in Equation 1 accordingly (N/mm2).

  1. Relevant sections of English version of KS F 4917 can be provided.

KS F 4917 has been largely brought over from the equivalent version of a Japanese standard from JIS A 6013, which in turn has been largely influenced by the ASTM versions of the tensile strength testing. The methods are essentially the same, but the variables for determining the degradation conditions vary due to the requirement to comply to the environmental conditions and regulations of the respective countries (for examples, ambient temp  erature conditions vary between the different national standards. The manuscript has also been revised such that the logic is no longer centering around the Korean standard, and proper references to ASTM, and BS EN standards have been provided as well. Please refer to the revised manuscript for details. We have included an English version of the KS F 4917, but please keep in mind that this standard is not to be reproduced or distributed in anyway.

Reviewer 2 Report

Dear Authors,

The article describes extensive research on composite materials joined with the use of five methods. While reading the manuscript, I had a comments that, hopefully, will help shape the final version of your article.

Manuscript is relatively long, but it describes a considerable number of research in detail.

I suggest you simplify the title of the article, it will positively affect its reading and recognition. The abstract should be supplemented with the most important quantitative results. I suggest choosing Keywords by searching for relevant terms in the search engine of MDPi publishing house.

The topic is well justified in the Introduction. Although Introduction is well prepared, citing only works and standards from one country does not mean that a broader review of the state of knowledge has been carried out. I suggest Table 1 to replace with figure and even two figures.

Figure 2 and 4: what does the word "status" mean? Test stand?

Line 219: correct the numbering of the subsection.

Chapter 3: Please consider whether the accuracy of the measurement methods justifies giving results with such high accuracy (3 decimal places). Maybe it results from calculating the average value?

Lines 295 and 334: information about the criterion was already given on line 267/268.

Note that in several places the font is partially invisible, e.g. lines 350, 372, above 432 ...

Figure 9: the font is too small.

References must be supplemented with current (last 3 years) articles from various countries and publishing houses.  Because it is impossible to verify the reliability of sources difficult to achieve, I suggest (where possible) to replace them for scientific articles.

Author Response

First and foremost, we would like to express our utmost gratitude for taking the time to read and comment on our manuscript. We understood from your feedback that our initial draft was rather difficult to understand due to poor quality in writing and arrangement. So we took the time to conduct further experimentation and revise the manuscript accordingly. Most importantly, we removed the latter part of the paper that relies on estimation for the results and replaced this section with experimental data on structural movement response testing (now called concrete displacement resistance) after environmental degradation. We hope the manuscript now reads more clearly and a complete review can be conducted. In the below, we have provided more specific response to your individual comments.

Dear Authors,

The article describes extensive research on composite materials joined with the use of five methods. While reading the manuscript, I had a comments that, hopefully, will help shape the final version of your article.

  1. Manuscript is relatively long, but it describes a considerable number of research in detail.

Response: The manuscript has been made shorter and succinct. We hope that the goal of the research at hand has been made more clear for the readers. Please refer to the revised manuscript (especially the introduction section for details).

  1. I suggest you simplify the title of the article, it will positively affect its reading and recognition.

Response: As the theme of this manuscript is rather complicated, we tried to encompass the whole topic into the title. While the title has not been shortened relatively from before,

  1. The abstract should be supplemented with the most important quantitative results.

I suggest choosing Keywords by searching for relevant terms in the search engine of MDPI publishing house.

Response:  Abstract has been revised to include the most relevant contents of the manuscript, and the key words have been revised accordingly (the MDPI publishing house does not specify keywords that may be relevant to the topic at hand, but they have been revised such that the terms are more general and applicable to the field of waterproofing more accurately)

  1. The topic is well justified in the Introduction. Although Introduction is well prepared, citing only works and standards from one country does not mean that a broader review of the state of knowledge has been carried out. I suggest Table 1 to replace with figure and even two figures.

Response:  Table 1 has been reformatted into a Figure format. References to international works and standards have been included and the focus on the Korean standards has been reduced to include a broader view (please refer to the new section 1.1.2 in the manuscript for details)

  1. Figure 2 and 4: what does the word "status" mean? Test stand?

Response:  it was an inappropriately translated expression from Korean. Figure 2 and 4 titles have been revised accordingly. 

  1. Line 219: correct the numbering of the subsection.

Response: numbering of the sections and subsections have been revised throughout the manuscript. Please refer to the revised version for details

  1. Chapter 3: Please consider whether the accuracy of the measurement methods justifies giving results with such high accuracy (3 decimal places). Maybe it results from calculating the average value?

Response: Data value presentation have been revised (all values have been fixed to two decimal places).

  1. Lines 295 and 334: information about the criterion was already given on line 267/268.

Response: Repetition on the information about the criterion has been removed in Lines 295 and 334

  1. Note that in several places the font is partially invisible, e.g. lines 350, 372, above 432 ...

Response: partially invisible fonts have been revised

  1. Figure 9: the font is too small.

Response: Figure 9 and the related section of the manuscript has been removed

References must be supplemented with current (last 3 years) articles from various countries and publishing houses.  Because it is impossible to verify the reliability of sources difficult to achieve, I suggest (where possible) to replace them for scientific articles.

Response: Some new recent references particularly to do with international standards have been added, but it must be noted that research on SCCW type of waterproofing membranes and interaction with concrete joint displacement while under the influence of environmental degradation is not a common topic that is discussed in the current field of studies (as SCCW is a new type of waterproofing system, and existing single-ply type sheet or coating materials have different types of problems with regards to installation, quality management, and degradation). Due to this reason, it was difficult to include articles from other countries on this topic. However, we expect that similar research topics will arise soon in East Asian countries and South East Asian countries in the coming years as this type of waterproofing system has been expanding in use in the recent years, but there is currently no precise evaluation methods aside from the ones used for singly-ply type waterproofing.

We hope that the revised version of the manuscript is easier to understand that it was before. Thank you kindly.

Reviewer 3 Report

The paper describes experimental results of great practical importance. Some tables with numbers could be simplified or presented in graphical form,  in order to make paper more readable and shorter.

Author Response

First and foremost, we would like to express our utmost gratitude for taking the time to read and comment on our manuscript. We took the time to conduct further experimentation and revise the manuscript accordingly. Most importantly, we removed the latter part of the paper that relies on estimation for the results and replaced this section with experimental data on structural movement response testing (now called concrete displacement resistance) after environmental degradation. 

Graphical versions of the tables have been added, and the latter section of the paper that relied mostly on tables have been removed. We hope that this revision contributes to an easier understanding of the paper.

Reviewer 4 Report

Dear Authors, 

The article is very interesting for researchers dealing with the subject of tensile performance of joints used for sheet-coating composite waterproofing.

The paper is very badly structured. I recommend rewriting the chapter Introduction. Moreover, some parts from other chapters should be transferred to Introduction as well.

I suggest completely revising the manuscript according to the comments that are included in the attached document and probably even more since it is really difficult to understand what was the main aim of the research and what are the most important findings.

With kind regards.

Author Response

First and foremost, we would like to express our utmost gratitude for taking the time to read and comment on our manuscript. We understood from your feedback that our initial draft was rather difficult to understand due to poor quality in writing and arrangement. So we took the time to conduct further experimentation and revise the manuscript accordingly. Most importantly, we removed the latter part of the paper that relies on estimation for the results and replaced this section with experimental data on structural movement response testing (now called concrete displacement resistance) after environmental degradation. We hope the manuscript now reads more clearly and a complete review can be conducted.

Dear Authors, 

The article is very interesting for researchers dealing with the subject of tensile performance of joints used for sheet-coating composite waterproofing.

The paper is very badly structured. I recommend rewriting the chapter Introduction. Moreover, some parts from other chapters should be transferred to Introduction as well.

I suggest completely revising the manuscript according to the comments that are included in the attached document and probably even more since it is really difficult to understand what was the main aim of the research and what are the most important findings.

With kind regards.

As you have included numerous comments and feedback to our paper, we will address some of the main and crucial points that you made in the hopes of contributing towards a clearer understanding of the paper.

  1. Abstract and introduction section have been substantially rewritten to improve the understanding of the goals of this research. Please refer to the revised manuscript for details.
  2. You mentioned in several sections about the importance of workmanship, and in the comment for line 52 and 53 you mention that material contributes only a small part of to the problem with regards to waterproofing quality. While this point is largely true, there is also a problem in the lack of understanding of the material properties that in turn leads to the initial problems about workmanship to begin with. Under laboratory setting, materials that seem to have high physical property are, in certain cases, very difficult to work with. In certain cases, despite having skilled workers, selection of improper materials (for example, using self-adhesive asphalt sheet for buildings located in regions with volatile climate conditions) leads to situations where high level workmanship is not enough to achieve high quality waterproofing work. The underlying point that is being discussed is that in the current state of affair, an improper evaluation of waterproofing materials is leading to difficulty in waterproofing construction (indirectly affecting workmanship). This is a very difficult topic to approach in a standard academic paper, as workmanship or quality of supervision is difficult to quantify, but must be addressed in some way or form. In Korea’s case, as this situation has been a prevailing problem in the last 20~30 years, recent trends in the field of waterproofing has been focusing strongly on developing high quality, but expensive waterproofing materials and techniques that would reduce the reliance on skilled workmanship (which leads to higher costs and longer time in construction). This trend has largely been successful thus far with the support of the government funding, but we are facing a different type of problem now, in that there is a new niche of waterproofing materials and techniques that require a fundamentally different approach for evaluation. This is precisely the case with SCCW type waterproofing and the research topic is to propose the requirement to develop a new type of evaluation system. As the introduction has been rewritten and the commented sections have been revised to make the above point more clearer, we hope that the manuscript will now read differently than before.
  3. The descriptions for the tensile strength testing and concrete displacement resistance testing has been added, and a clearer explanation as to why a correlative analysis on the results of the two types of testing methods is required has been provided.
  4. With regards to your point on needing a more globalized reference, research into international standards has been added. However, with regards to the actual topic itself (on composite waterproofing membrane joint section), research on this field is actually quite scarce. As you have mentioned there are many research conducted on singly-ply and multi-ply type waterproof roofing (particularly in Japan and China), but it is difficult to find a published work that concerns the same situation that is being address in the manuscript. We still have included a few new references, but we hope you can understand the reasoning as to why only a few have been newly added.

Other minor comments and feed backs have been duly noted and we hope that the revised version of the manuscript is easier to understand that it was before. Thank you kindly.

Round 2

Author Response

Thank you kindly for taking your time to look over the manuscript once again. Your attentive care and your comments have been paramount to the drafting of this manuscript, and it has been significantly improved thanks to your contribution.

We have provided the following report to your comments for the minor revision;

  1. SCCW should be renamed Sheet-Coated Composite Waterproofing instead of Sheet-Coating in the Abstract and introduction section throughout. 

Response: SCCW has been renamed to “sheet-coated composite waterproofing” throughout the paper (one was revised in the introduction section)

  1. In Table 1, the ratio ?/A is longitudinal strain. The term 1 + a/A has no name in Physics.

Response: The explanation on the variables in Table 1 has been revised and made simpler for clearer understanding

  1. Comments in Table 3, megapascals is written as ???, no ???.

Response: Mpa has been revised to a correct acronym.

  1. Line 738: What is coefficient of change?

Response: This concept has been removed from the paper, but it seems we forgot to remove it from that particular line in the previous revision.

  1. Line 754: What is ?/? as defined in KS?

Response: W/C is water to cement ratio for making the mortar substrate part during the concrete displacement testing. This expression in Line 754 has been rewritten to make the definition more clear

  1. In Tables 7 and 8,and Fig 10, the measure of tensile strength should be indicated in ?/??2 (=???).

Response: Tensile strength units in Tables 7, 8 and Fig.10 have been revised accordingly

  1. In Fig. 11, what are the units of the number plotted on the vertical scale?

Response: The vertical scale does not have a scientific unit, and it is measure of the interval at which leakage occurred during the concrete displacement resistance testing (since every cycle consists of 100 intervals of movement, we expressed the vertical scale as such). We revised the vertical scale label in Fig. 11 to make the meaning more clear.

Once again, we would like to express our sincerest gratitude for your review of our manuscript.

Reviewer 4 Report

Dear, 

there are still some flaws but overall, the article is very interesting and the experiment is described in a better way.

Kind regards.

Author Response

Thank you kindly for all your attentive care and comments for our manuscript. Your review has been a tremendous help in improving the quality of our paper. 

We made a few technical revisions to our paper so that the results are more scientifically correct, and we added a few lines at the end of Section 3 to illustrate that more research will be required in the future to ensure that the method proposed in this research paper will more objective. While for the moment the issue in this paper seems limited in application in a global scale, we are certain based on the trends of development in waterproofing technology that a new evaluation method (more improved than the model demonstrated in this paper) will need to be established for new materials/techniques such as the SCCW systems. 

We would like to express our sincerest gratitude to you once again for all your contribution and assistance in the drafting of this manuscript